# What We Know So Far About ECG for Pancreatic Pseudocysts

**DOI:** 10.3390/life14111419

**Published:** 2024-11-04

**Authors:** Paulina Kluszczyk, Beata Jabłońska, Michał Serafin, Aleksandra Tobiasz, Tomasz Kowalczyk, Sebastian Maślanka, Mateusz Chapuła, Piotr Wosiewicz, Sławomir Mrowiec

**Affiliations:** 1Student Scientific Society, Department of Digestive Tract Surgery, Faculty of Medical Sciences in Katowice, Medical University of Silesia, Medyków 14 St., 40-752 Katowice, Poland; kluszczyk.paulina@gmail.com (P.K.); michal.serafin@interia.com (M.S.); olatob@gmail.com (A.T.); tomekk2601@gmail.com (T.K.); s.maslanka98@gmail.com (S.M.); 2Department of Digestive Tract Surgery, Faculty of Medical Sciences in Katowice, Medical University of Silesia, Medyków 14 St., 40-752 Katowice, Poland; mrowasm@poczta.onet.pl; 3Department of Gastroenterology and Hepatology, Faculty of Medical Sciences in Katowice, Medical University of Silesia, Medyków 14 St., 40-752 Katowice, Poland; mchapula@uck.katowice.pl (M.C.); doctorw@poczta.onet.pl (P.W.)

**Keywords:** post-inflammatory pancreatic cysts, endoscopic internal drainage, endoscopic cystogastrostomy

## Abstract

**Background:** Endoscopic cysto-gastrostomy (ECG) has become the treatment of choice for pancreatic pseudocysts (PPCs). Endoscopic drainage of PPCs requires the creation of an anastomosis between the lumen of the PPCs and the lumen of the gastrointestinal tract. Various types of stents are used for this purpose. The aim of the study is to compare the indications, quantity, and results of using double pigtail plastic stents (DPPSs) and lumen-apposing fully covered metal stents (LAMSs) in ECG. **Methods:** A retrospective analysis was conducted of 39 patients (24 men, 15 women) treated for PPCs in the Department of Digestive Tract Surgery and the Department of Gastroenterology and Hepatology between October 2018 and February 2023. The mean age of patients was 51.13 (28–77). Data about etiology, cyst diameter, type, and complications of the stents were collected. **Results:** DPPSs were placed in smaller cysts (108 vs. 140 millimeters, *p* = 0.04) and were maintained for a longer duration compared to LAMSs (106 vs. 34 days, *p* = 0.001). Cyst recurrence was reported less frequently in patients with a LAMS (0 (0%) vs. 4 (19.05%), *p* = 0.05) and the therapeutic success was non-significantly higher in the LAMS group compared to the DPPS group (100% vs. 85.71%), *p* = 0.095. **Conclusions:** Both DPPSs and LAMSs are characterized by high therapeutic success and low complication rates in patients undergoing ECG for PPCs.

## 1. Introduction

Pancreatic pseudocysts (PPCs) are fluid collections containing pancreatic juice, rich in amylase and other pancreatic enzymes [1,2]. These cysts are enclosed by a wall of connective tissue surrounding the pancreas. According to the revised Atlanta Classification (2012), PPCs are defined as well-defined, encapsulated fluid collections with no or minimal solid components that develop more than four weeks after the onset of interstitial edematous pancreatitis [1,3,4]. In adults, the incidence of PPCs is estimated to be between 1.6% and 4.5%, with an occurrence rate of 0.5 to 1 per 100,000 adults annually. The primary causes of PPCs include acute pancreatitis (AP), pancreatic injury, and chronic pancreatitis (CP). The incidence of PPCs in AP ranges from 6% to 18.5%, while in CP it ranges from 20% to 40% [5,6].

Pseudocysts are often asymptomatic. However, when symptoms occur, the most common include abdominal pain, nausea, vomiting, abdominal bloating, and the presence of an abdominal mass. Complications primarily arise from the compression of the duodenum, common bile duct, or pylorus, leading to the narrowing of these lumens. Additionally, infection of the cyst or bleeding into the lumen of a PPC are severe and potentially life-threatening complications [7,8,9].

About two-thirds of cysts can be managed by active observation. The remainder of patients require cyst drainage [10]. In such cases, it is necessary to provide appropriate treatment methods, taking into account parameters such as size, location, type of cyst, and clinical status of the patients [11]. It is very important to differentiate the pseudocyst from a variety of cystic tumors. Ultrasonography, computer tomography (CT) of the abdomen, endoscopic ultrasonography (EUS), and biopsy are preferred in the diagnosis of PPCs [12]. The two main indications for intervention are symptomatic PPCs or presence of complications such as infected pseudocyst, digestive tract obstruction, pancreatic portal hypertension, biliary obstruction, obstructed gastric outlet, bleeding, and others [2,10]. It is worth noting that the size of the cyst alone is not an indication for intervention [13,14]. 

Historically, pancreatic cysts have been treated using a variety of techniques, including internal drainage methods such as cystogastrostomy, cystoduodenostomy, and cystojejunostomy; external percutaneous drainage; marsupialization; and pancreatic resection [8,15,16,17].

However, in recent years internal drainage and more specifically, endoscopic cysto-gastrostomy (ECG) has become the treatment of choice for PPCs [18,19]. In recent years, the focus in endoscopic management of pancreatic cysts has shifted towards evaluating the efficacy of different stent types used in drainage procedures. Two types of stents are currently employed: double pigtail plastic stents (DPPSs) and lumen-apposing fully covered metal stents (LAMSs), a type of self-expanding metal stent (SEMS) [20]. LAMSs offer a larger diameter compared to plastic stents, which theoretically allows for more efficient and rapid drainage [21]. Therefore, LAMSs are used in the endoscopic treatment of both PPCs and walled-off necrosis (WON). In WON, LAMSs allow direct endoscopic necrosectomy (DEN) [18]. However, some studies have found no significant difference in procedural effectiveness between plastic and metal stents [21]. Metal stents are associated with more recurrences and late complications such as bleeding, buried stents, and biliary stricture [21,22,23], suggesting that they may require earlier removal. The debate over the superiority of metal versus plastic stents remains unresolved, with some studies supporting the advantages of metal stents [23,24,25,26], while others do not [22,27,28,29].

Thus, the primary objective of this retrospective analysis was to compare the short-term and long-term outcomes of endoscopic cystogastrostomy using plastic and self-expanding metal stents based on data obtained from the Department of Digestive Tract Surgery and the Department of Gastroenterology and Hepatology, at the Medical University of Silesia in Katowice, Poland.

## 2. Materials and Methods

### 2.1. Study Design and Patient Cohort

The retrospective study included adult patients from the Department of Digestive Tract Surgery and the Department of Gastroenterology and Hepatology, Medical University of Silesia in Katowice, Poland, who were treated with ECG for PPCs from January 2018 to February 2023. 

There were 24 (61.54%) men and 15 (38.46%) with average age 51.13 (28–77, SD 13.18) in the analyzed group. The inclusion criteria were as follows: age ≥ 18 years, PPC caused by AP or CP confirmed by abdominal computed tomography, ECG as the treatment method, and availability of medical records for the data analyzed. The exclusion criteria were as follows: age < 18 years, true or complicated (hemorrhagic, infected) pancreatic cyst, conservative or other invasive PCC treatment, and incomplete demographic and/or clinical data. 

#### Inclusion Criteria for ECG

An ECG was conducted for patients with PPCs that demonstrated a well-defined wall and presented with symptoms or complications. The distance between the PPC wall and the gastric wall was ≤1 cm, indicating a significant impression on the stomach, which facilitated the endoscopic approach. Furthermore, patients selected for this procedure had no contraindications for endoscopic interventions, such as coagulopathy, severe comorbid conditions, or active infection that could complicate the procedure. 

### 2.2. Methods of Treatment

All patients were evaluated by surgeons and/or gastroenterologists, who decided on conservative treatment (monitoring) or qualification for endoscopic drainage. All procedures were performed by experienced endoscopists. 

### 2.3. Endoscopic Cysto-Gastrostomy (ECG)

ECG is a direct drainage through the wall of the stomach or duodenum using a metal or plastic prosthesis under EUS and fluoroscopic guidance. The ECG procedure is performed using the Olympus UTC 180 endoscopic ultrasound scope (Tokyo, Japan) under intravenous procedural sedation. The endosonograph was introduced through the oral cavity into the stomach, where, under ultrasound control, the PPC was visualized. The next step was to identify the optimal site and puncture the cyst with an Olympus 19G needle or Cook Cystotome™ (Bloomington, IN, USA). The cyst fluid was collected for biochemical evaluation. A guidewire was passed through into the lumen of the cyst, and under fluoroscopy control, the puncture was dilated using rigid dilators of 6 to 10 F or with the balloon catheter. With the guidance of the guidewire, a stent (DPPS or LAMS) was introduced, allowing the fluid content of the cyst to drain into the gastrointestinal tract (Figure 1). 

The drainage mechanism in the LAMS consists in creating a wide (14–16 mm) channel between the fluid collection and the stomach lumen, which allows for quick drainage of the contents and, if necessary, endoscopic necrosectomy. In the case of DPT, the mechanism is complex and initially consists in drainage through the prosthesis lumen (specifically through the main channel and small holes in the prosthesis wall), and after creating a fistula channel also around the prosthesis. Inserting two prostheses through one channel connecting the collection with the stomach lumen accelerates drainage by increasing the flow through the prostheses and around them.

During the endoscopic ultrasound-guided drainage (EUS), a DPPS (size 7F or 10F) or a Hanarostent^®^ Plumber™ LAMS (size 10/12/14/16 mm) from M.I.Tech (Pyeongtaek-si, Gyeonggi-do, Republic of Korea) was inserted. 

A LAMS is a type of metal, fully covered, removable, self-expandable stent (self-expandable metallic stent, also called self-expandable metal stent) with wide flanges at the ends, which limits the possibility of dislocation of the stent. In our Endoscopy Unit, we use Olympus HANAROSTENT^®^ Plumber™ with a diameter of 14 or 16 mm and a length of 30 or 40 mm.

The choice of stent was made by the experienced endoscopist performing the procedure, based on the patient’s clinical status, the size, and the localization of the PPC, and the contents of the PPC as assessed by EUS.

Initially, we used two sizes of LAMS—14 and 16 mm in diameter and 30 and 40 mm in length. Currently, we more often use 16 mm lumen diameter stents (they facilitate necrosectomy and revision of the fluid collection) and the length of the prosthesis depends only and exclusively on the distance of the pseudocyst from the wall. In the case of plastic DPD stents, we usually use one with a diameter of 10 F (3.33 mm). The selection of the length of the DPT depends on the size of the collection.

### 2.4. Analyzed Data

The study analyzed parameters including cyst localization, diameter, etiology, type of clinical symptoms, American Society of Anesthesiologists (ASA) score, surgery duration, incidence of complications, reinterventions, mortality, hospitalization duration, and the types of stents used in the endoscopic drainage.

General characteristics of the patients, such as age, gender, and clinical symptoms, as well as the incidence of postprocedural complications, and rehospitalizations, were extracted from the patient’s medical records. 

Information on the procedure duration and the types of stents used was obtained from the procedural descriptions. Cyst localization and diameter were determined from patient CT scans (Figure 2). Follow-up data, including complications and late reinterventions, were gathered from the patient’s medical history in the surgical clinic and/or the department.

The studied cohort was divided into two groups according to the type of stent used during the ECG (DPPS and LAMS groups).

### 2.5. Statistical Analysis

Statistical analysis was carried out using Statistica^®^ software (version 13.3, StatSoft, Polska, Kraków, Poland). Qualitative variables were expressed as absolute values and percentages, whereas quantitative variables were presented as ranges, means, and standard deviations, or medians with interquartile ranges. The Shapiro–Wilk test was used to assess the statistical distribution of the patients.

Comparisons between plastic stent and metal stent groups were made regarding demographics, periprocedural and postprocedural parameters, and follow-up data, employing the chi-square test, Fisher’s exact test, Student T-test, or Mann–Whitney U test. Predictive factors for early postprocedural complications were determined using univariate logistic regression analysis.

Freedom from cyst recurrence was evaluated using the Kaplan–Meier estimator, with the log-rank test employed for comparisons between the stent types. Statistical significance was set at a *p*-value of <0.05.

### 2.6. Definitions 

Therapeutic success was defined as the postprocedural complete resolution of the pseudocyst or a significant decrease (to <3 cm in diameter) in the imaging method with an overall improvement in symptoms after the first intervention. 

Treatment was considered to have failed in patients who required a repeat intervention. 

Recurrence was defined as a new pseudocyst observed by imaging methods at follow-up after a previously reported resolution. Moreover, in the meantime, there was no further AP or exacerbation of CP. 

Total time of hospitalization was the length of the stay from the day of the hospital admission to the discharge. The time of the hospitalization after the procedure was the length of stay from the day of endoscopic approach to discharge.

Follow-up was defined as the patient’s last registered visit to the ward or clinic. 

## 3. Results

### 3.1. Patients’ Characteristics

There were no statistically significant differences in the distribution of age, sex, or body mass index (BMI) between the compared patient groups.

The most common symptoms were abdominal pain observed in 28 (71.79%), followed by weight loss in 10 (25.64%) patients. There were no statistically significant differences in the symptoms between the two groups.

Most patients (28; 71.79%) were assigned to group II of the American Society of Anesthesiologists (ASA) score. There were no differences in the ASA assessment among patients with plastic and metal stents. Demographic and clinical characteristics of patients compared according to stent type are presented in Table 1.

### 3.2. Cyst Characteristics

The most common location of the cysts was the pancreatic tail in patients undergoing ECG with LAMSs (*n* = 7; 38.89%), and the pancreatic body and tail in patients undergoing ECG with DPPSs (*n* = 6; 28.57%). There were no differences between the two groups concerning cyst locations. 

In both compared groups, the most common etiology of cysts was AP (*n* = 14; 66.67%) in the DPPS group and (*n* = 13; 72.22%) in the LAMS group, (*p* = 1). Other cysts resulted from CP or had a history of both AP and CP. There were no differences between the two groups regarding the etiology. 

In comparison, plastic stents were placed in smaller cysts (108 vs. 140 millimeters, *p* = 0.04). All mentioned above comparisons are presented in Table 2.

### 3.3. Short-Term Outcomes

The duration of procedure for insertion of the DPPSs was shorter than the LAMSs (35 vs. 50 min, *p* < 0.001). Duration of hospitalization after the primary procedure was longer in the LAMS group compared to DPPS group (8 vs. 3 days, *p* = 0.002). The duration of hospitalization was similar in both groups (*p* = 0.07).

A total of 5 (12.82%) complications occurred, including 3 (14.29%) following DPPS insertion and 2 (11.11%) following LAMS insertion. The frequency and types of complications were similar in both groups (*p* = 0.77). In both compared groups, the 30-day mortality was 0 (0%). 

There was no significant difference in the number of reinterventions (*p* = 0.76) and rehospitalizations (*p* = 0.30) as well as in median time of follow-up in days (*p* = 0.34).

Cyst recurrence was reported less frequently in patients with LAMSs compared to DPPSs (0 (0%) vs. 4 (19.05%), *p* = 0.05). Therapeutic success was higher in the LAMS group compared to the DPPS group (100% vs. 85.71%). Nonetheless, this was not statistically significant (*p* = 0.09). In 3 (14.29%) patients with plastic stents without therapeutic success, an additional metal stent was used.

The plastic stents were maintained for a longer duration compared to metal stents (106 vs. 34 days, *p* = 0.001). All complications and follow-up data are presented in Table 3. 

The median duration of the procedure decreased from 75 min in 2018 to 35 min in 2023 in the LAMS group and from 67.5 min in 2019 to 25 min in 2023 in the DPPS group (Figure 3).

In univariate regression analysis, none of the variables proved to be an independent predictive factor for postprocedural complications (Figure 4).

The 6-month freedom from cyst recurrence was 91.17% (standard error (SE) = 4.86%). There was no statistical significant difference in the 6-month freedom from cyst recurrence between DPPS and LAMS groups (85.00% SE = 7.98% vs. 100% SE 0%; *p* = 0.07) (Figure 5).

In univariate analysis, only occurrence of postprocedural complications proved to be an independent predictive factor of cyst recurrence (hazard ratio = 21.55, 95% CI = 1.95–237.72, *p* = 0.01) (Table 4).

## 4. Discussion

PPCs are unpredictable. They may occur asymptomatically or present with a variety of symptoms such as abdominal pain, nausea, vomiting, early satiety, and upper gastrointestinal bleeding. Most pancreatic pseudocysts are managed without an intervention, by observation during which the cysts usually resolve spontaneously [2,10,30]. After the maturation period of PPCs, which lasts 2–6 weeks, only one-third of them are not resolved spontaneously. These should be considered for invasive treatment [31]. The size at which a cyst requires intervention is under discussion because, as mentioned above, size alone is not an indicator for drainage [2,13,14]. The two main indications allowing one to qualify a patient for drainage procedures are symptomatic pseudocysts and/or the presence of some complications like infected pseudocyst, digestive tract obstruction, portal hypertension, biliary obstruction, obstructed gastric outlet, bleeding, and others [2,10]. Moreover, studies have indicated that the length of time the cyst has been present is not a good enough predictor of potential resolution or complication of pseudocyst [2,32]. Therefore, one study suggests that patients with cysts >3 cm should be referred for surgical consultation; however, usually invasive treatment is not conducted without other symptoms and when the size of the cyst is smaller than 4 cm [33]. In recent studies, the size of the cysts treated with internal drainage ranged between 6 cm and 40 cm [10,14,34]. In our study, cysts of similar size were treated invasively, raging between 6.5 cm and 21.6 cm.

Pancreatic pseudocysts can be treated with a variety of methods like endoscopic, percutaneous, and surgical drainage. Currently, endoscopic treatment is the treatment of choice for PPCs due to its less invasive approach and high long-term success rate, but still the optimal treatment plan should be determined according to the different conditions of the patient [2,10]. The first endoscopic transpapillary drainage was described in 1984 and one year later the first pancreatocystogastroduodenostomy (internal cyst drainage into the stomach or duodenum) was performed [35,36,37]. Initially, it was used for drainage of pancreatic cysts connecting to the Wirsung’s duct during endoscopic retrograde cholangiopancreatography (ERCP). However, this solution was replaced over time with transmural drainage, which consists of the evacuation of the cyst fluid to the gastrointestinal tract using a metal or a pig-tail plastic prosthesis. [37]. In a systematic review, Farias et al. have shown an increasing trend in the number of endoscopic drainages compared to surgical ones performed since 2008 [38]. The endoscopic method is less invasive than surgery and avoids the need for external drainage, yet the target population is limited because of the requirements with regard to cyst location. Studies have shown that the distance between the cyst wall and the gastric or duodenal wall must be less than 1 cm [2,10,39,40,41]. A relative contraindication for endoscopic drainage is the presence of large intertwined vessels or varices [40,41]. The feasibility of stenting of a pseudocyst is ensured by fluoroscopic guidance or EUS, which is used to detect an optimal topography of the pseudocyst, intestinal wall, and interfering vascular system. One study shows that a higher risk of failure to identify an appropriate site using EUS occurred when pseudocysts were located in the area of the pancreatic tail or had a less prominent endoscopic impression [2,34]. Additionally, EUS is also beneficial in reducing the incidence of iatrogenic hemorrhage [42].

Nowadays, the major area of interest in endoscopy is the type of stent used in drainage. We can distinguish two types of the stent: DPPSs and LAMSs [20]. There are several differences between them. LAMSs provide larger diameter compared to DPPSs. In most cases, LAMSs have >10 mm compared with DPPSs (7-or 10-Fr.; 2.3 or 3.3 mm). Therefore, they should ensure better and faster drainage than plastic stents and therapeutic success, which is confirmed by several studies [21,23,43]. Plastic stents can often become occluded, thus causing a restriction of fistula drainage and requiring stent replacement. A major factor that determines the drainage time and occlusion of these stents is the pseudocyst cavity [21]. Nevertheless, in our study there was no difference in effectiveness of the procedure between plastic and metal stents, 18 (85.71%) vs. 18 (100%); *p* = 0.095. Our results are similar to most authors’ reports, because according to the huge systematic review, treatment success rate in both metal and plastic stents is high (>80%) [43] or even higher (>90%) [23,44]. However, in one meta-analysis authors present a significantly higher clinical success rate and lower overall adverse event rate of metal stents compared to plastic stents for pseudocysts and walled-off necrosis [23].

The choice of the type of prosthesis depends mainly on the presence of solid masses inside the fluid collection. The second important criterion is the presence of a fistula between the collection and the pancreatic duct. In the first case, the placement of a metal stent allows for more effective drainage. The presence of a pancreatic fistula may require long-term maintenance of a drainage prosthesis, which in the case of metal stents is associated with a higher risk of bleeding [22].

Regarding the selection of the appropriate type of prosthesis in relation to the size of the cyst present, one study shows no relationship in this field, since the average size of the cyst is the same for both of the groups [24] and another shows the same result comparing the group with a plastic stent and metal with selective application of plastic stents [45]. Other studies only describe the mean average cyst size (102 mm) of the entire study group [23,44]. However, in our study, plastic stents were placed in smaller cysts (108 vs. 140 mm, *p* = 0.04). 

Nonetheless, in the literature, it is mentioned that the initial pseudocyst size has no impact on the success or failure of the drainage [21]. This difference is probably associated with the fact that larger cysts in our study were denser. 

However, the choice of prosthesis is also influenced by the number of drained fluid collections. In patients with two or more, it seems reasonable to deploy DPT plastic stents into each of them. The insertion of several metal prostheses could be too traumatic, as well as very expensive. It must be admitted that, considering the high and comparable drainage effectiveness of the LAMS and DPD techniques, the choice of one of them is often a subjective decision of the endoscopist [22].

The mean duration of the ECG was significantly higher in the metal stent group compared to the plastic stent group (50 vs. 35 min, *p* < 0.001), but this result is not confirmed by any study. In each of the three studies subjected to meta-analysis, the median or mean duration of the procedure with metal stents was significantly shorter than with plastic stents [23]. Nonetheless, in our study, the median duration of the procedure shortened from 75 min in 2018 to 35 min in the LAMS group and from 67.5 min in 2019 to 25 min in 2023 in the DPPS group. Initially, due to the lack of a cystotome, all procedures required needle puncture, guidewire insertion, and dilation with a rigid plastic dilator or a balloon, which significantly extended the procedure time. However, with the acquisition of a cystotome, the duration of these procedures was substantially reduced. Additionally, the longer duration of procedures in the LAMS group compared to the DPPS group may be attributed to the fact that LAMS devices were only introduced in 2018 at our department. Consequently, the period between 2018 and 2020 represented a learning phase for the implantation of LAMSs, during which the procedure initially took longer. Over time, as proficiency increased, the duration of the procedure was reduced.

Moreover, our study revealed that there was a statistical difference in the duration of hospitalization after ECG between DPPS and LAMS groups (3 vs. 8 days). A few detailed analyses of the time of hospitalization after the procedure does not allow us to compare our results. One study shows only a similar length of hospitalization for both groups, which is mean about 8 days, while the median time in the group of plastic stents is shorter than in the group of metal with selective application of plastic stents (2 vs. 4 days) [45].

Complications following ECG include gastrointestinal and intra-abdominal bleeding, infection, digestive tract perforation, stent migration, and septic shock caused by cyst infection due to inadequate drainage [18,46]. Patients treated with LAMSs have lower complications rates [44]. In our group, postprocedural complications occurred in three (14.29%) patients treated with plastic stents and in two (11.11%) patients with metal ones (*p* > 0.05). Postprocedural 30-day mortality was 0% in both groups. These data are comparable with the literature showing complications in 3.3–36.1% in ECG with plastic stent implantation and 0–23.7% with implantation of metal stents; it is noteworthy that the total number of adverse events after ECG was significantly higher in the plastic than in the metal stent group [23,24].

Data on adverse events between drainage with metal and plastic stents remain varied. Late complications associated with the use of stents such as bleeding, buried stents, stent migration, stent obstruction, and biliary stricture were reported in some studies [21,22,23,47]. 

Bleeding after stent placement and stent migration occurs with similar frequency, but one study shows that bleeding (including live threatening bleeding) is more common in the LAMS group of patients [44]. The risk of bleeding, which can occur at any stage of the procedure, should therefore not determine the method of drainage. There is also no significant difference in time to bleeding [23,47].

Migration of the DPT plastic stent occurs slightly more often than the LAMSs [44]. The risk of metal stent migration increases with time, but is no longer a problem since the introduction of metal, self-expanding, fully covered LAMS prostheses [44,47]. However, due to the lack of any anchorage force, they are not a good option in cases when the cyst is not firmly attached to the gastric wall, because the risk of leakage is high [48]. Thus, metal stents should be removed faster—within 4 weeks—than plastic stents, which can be left indwelling [22,45]. In our group, the median removal time was 106 days for plastic stents compared to 34 days for metal stents (*p* = 0.001); that is similar to the literature data [23]. 

The recurrence rate is between 3.4% and 15% in the plastic stent group and 0.9–15% in the metal stent group and there was no statistical difference [23,43]. According to our results, cyst recurrence occurred in 19.05% of patients with plastic stents and in 0% in the metal stent group (*p* = 0.05). In both groups, the 6-month freedom from cyst recurrence was 91.17%; in another study, it was 90.4%, which was confirmed by CT scan [24].

The lower recurrence rate can be associated with the fact that LAMSs have larger diameter (in most cases > 10 mm) compared with DPPSs (7-or 10-Fr.; 2.3 or 3.3 mm). Therefore, the larger diameter prevents stent occlusion (associated with a higher risk of infectious complications and the need to replace or add an additional stent) and cyst recurrence [23,44]. 

The superiority of metal stents is still uncertain and is suggested by some studies [23,24,25,26] yet in others it is not demonstrated [22,27,28,29]. Their usage should depend on the individual assessment of the endoscopist.

### Limitations

One limitation of this study is that it is a retrospective analysis conducted on a selected cohort of patients from a single high-volume center. Another limitation is the relatively small sample size. In addition, the use of plastic or metal stents was determined by the endoscopist. Additionally, the restricted access to hospitals during the COVID-19 pandemic could have affected the timing of prosthesis removal, potentially distorting the data on prosthesis-retention times. Lastly, our follow-up period was limited, preventing us from fully capturing longer-term outcomes related to prosthesis retention and patient health. To address these limitations, future studies should consider a prospective, randomized, multicenter design with larger, more diverse patient populations. 

## 5. Conclusions

The majority of PPCs managed endoscopically are symptomatic and abdominal pain is the most common clinical symptom. Two types of stents (DPPSs and LAMSs) are used in ECG. Metal stents, with their larger diameter, offer advantages such as better drainage and lower recurrence rates compared to plastic stents. DPPSs are used more frequently in smaller cysts and are maintained for a longer duration compared to LAMSs. The cyst recurrence is reported less frequently in patients with LAMSs and the therapeutic success is non-significantly higher in the LAMS group compared to the DPPS group. Generally, both stent types used in ECG are characterized by high therapeutic success and low complications rate. Therefore, the choice between stents should be individualized and performed by an experienced endoscopist. 

## Figures and Tables

**Figure 1 life-14-01419-f001:**
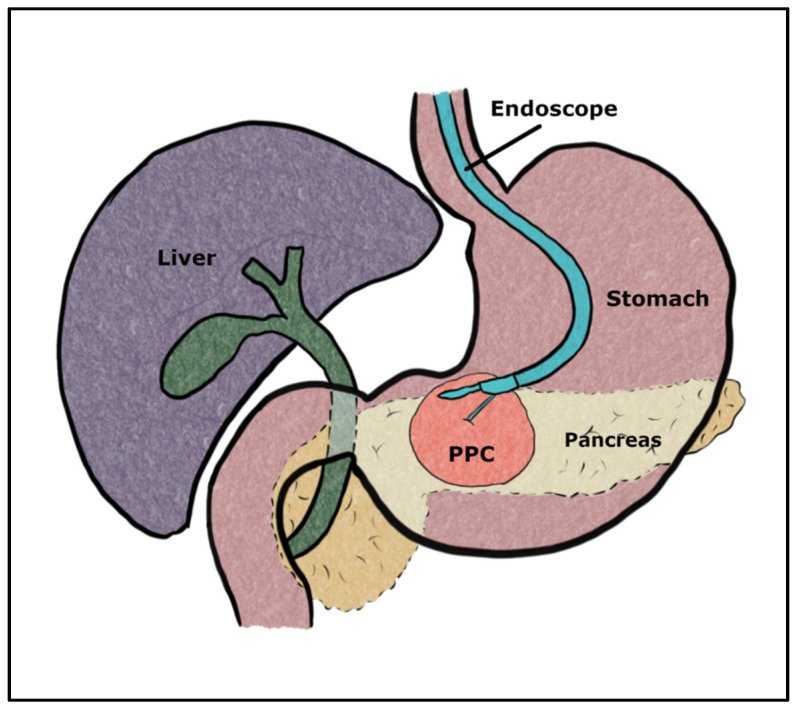
Schematic presentation of endoscopic cystogastrostomy procedure.

**Figure 2 life-14-01419-f002:**
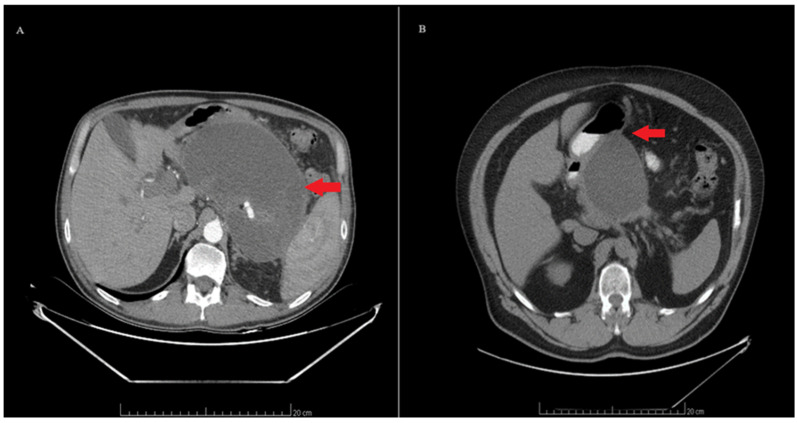
Computed tomography: (**A**) large pancreatic pseudocyst; (**B**) pancreatic pseudocysts with an impression on the stomach wall.

**Figure 3 life-14-01419-f003:**
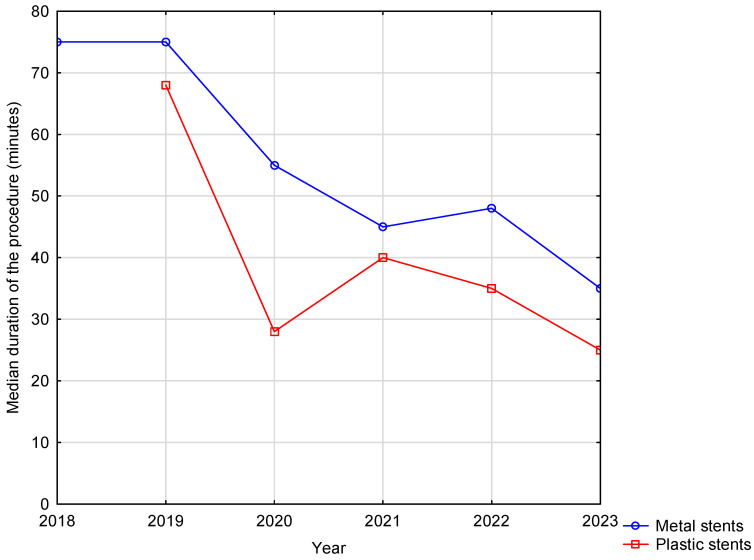
Median duration of the procedure depending on the type of the procedure and year.

**Figure 4 life-14-01419-f004:**
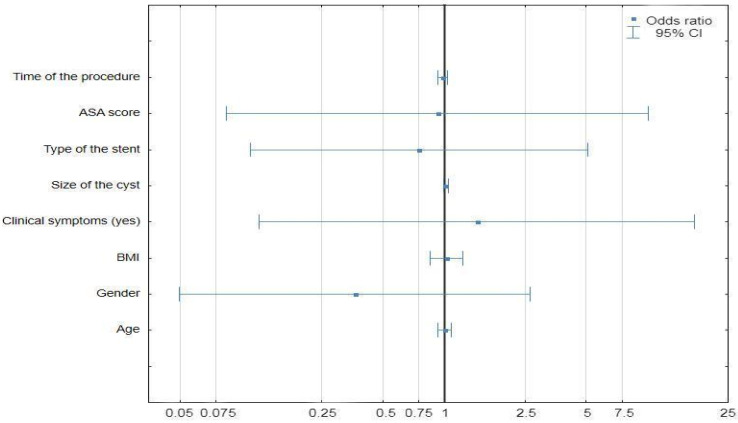
Univariate logistic regression analysis of predictive factors of postprocedural complications (Statistica^®^, 13.3, StatSoft). Abbreviations: 95% CI—95% Confidence Interval.

**Figure 5 life-14-01419-f005:**
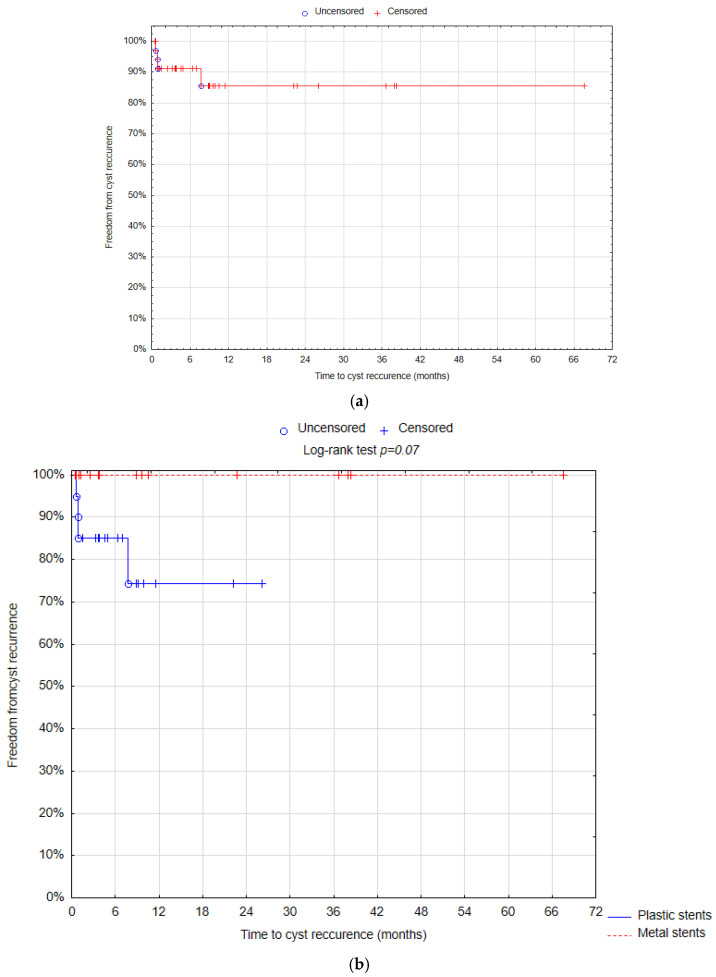
Freedom from cyst recurrence after 6 months—overall (**a**); DPPS vs. LAMS group (**b**) (Statistica^®^, 13.3, StatSoft).

**Table 1 life-14-01419-t001:** Demographic and clinical characteristics of patients.

	DPPS (*n* = 21, 53.85%)	LAMS (*n* = 18, 46.15%)	Total	*p*
Age (years)	53.45 SD 13.55 (30–70)	48.28 SD 12.49 (28–65)	51.13 (28–77, SD 13.18	0.23
Gender				0.74
Male	12 (57.14%)	12 (66.67%)	24 (61.54%)	
Female	9 (42.86%)	6 (33.33%)	15 (38.46%)	
Body Mass Index (BMI)	24.29 SD 5.62 (14.90–34.93)	24.51 SD 4.97 (15.42–36.41)	24.40 SD 5.25 (14.90–36.41)	0.90
Symptoms				
Abdominal pain	17 (80.95%)	11 (61.11%)	28 (71.79%)	0.28
Nausea and vomiting	4 (19.05%)	3 (16.67%)	7 (17.95%)	1
Jaundice	0 (0%)	1 (5.56%)	1 (2.56%)	0.27
Fever	2 (9.52%)	0 (0%)	2 (5.13%)	0.49
Steatorrhea	1 (4.76%)	0 (0%)	1 (2.56%)	1
Constipation	1 (4.76%)	3 (16.67%)	4 (10.26%)	0.22
Diarrhea	0 (0%)	1 (5.56%)	1 (2.56%)	0.27
Weight loss	5 (23.81%)	5 (27.78%)	10 (25.64%)	1
None	3 (14.29%)	3 (16.67%)	6 (15.38%)	1
American Society of Anesthesiologists (ASA) score				
I	0 (0%)	0 (0%)	0 (0%)	1
II	16 (76.19%)	13 (72.22%)	28 (71.79%)	1
III	5 (23.81%)	5 (27.78%)	10 (25.64%)	1
IV	0 (0%)	0 (0%)	0 (0%)	1

Abbreviations: DPPSs, double plastic pigtail stents; LAMSs, lumen-apposing fully covered metal stents; SD—Standard Deviation.

**Table 2 life-14-01419-t002:** Characteristics of pancreatic cysts.

	DPPS (*n* = 21, 53.85%)	LAMS (*n* = 18, 46.15%)	Total	*p*
Localization of the cyst				
Head of the pancreas	4 (19.05%)	4 (22.22%)	8 (20.51%)	0.81
Body of the pancreas	2 (9.52%)	1 (5.56%)	3 (7.69%)	0.64
Tail of the pancreas	6 (28.57%)	7 (38.89%)	13 (33.33%)	0.49
Head and body of the pancreas	3 (14.29%)	0 (0%)	3 (7.69%)	0.09
Body and tail of the pancreas	6 (28.57%)	6 (33.33%)	12 (30.77%)	0.75
The greatest size of the cyst	108 IQR 23 (65–216)	140 IQR 30 (71–200)	120 IQR 45 (65–216)	0.04
Cyst etiology				
Acute pancreatitis	14 (66.67%)	13 (72.22%)	27 (69.23%)	1
Chronic pancreatitis	6 (28.57%)	2 (11.11%)	8 (20.51%)	0.18
Both acute and chronic pancreatitis	1 (4.76%)	3 (16.67%)	4 (10.26%)	0.22

Abbreviations: DPPSs, double plastic pigtail stents; LAMSs, lumen-apposing fully covered metal stents; IQR—Interquartile Range.

**Table 3 life-14-01419-t003:** Short-term and long-term outcomes.

	DPPS (*n* = 21, 53.85%)	LAMS (*n* = 18, 46.15%)	Total	*p*
Duration of procedure (minutes)	35 IQR 15 (15–50)	50 IQR 30 (30–75)	40 IQR 30 (15–75)	<0.001
Duration of hospitalization (days)	5 IQR 8 (2–36)	11 IQR 11(3–31)	10 IQR 13 (2–31)	0.07
Duration of hospitalization after procedure (days)	3 IQR 4 (1–25)	8 IQR 8 (2–27)	5 IQR 8 (1–27)	0.002
Complications				
Total	3 (14.29%)	2 (11.11%)	5 (12.82%)	0.77
Bleeding into the cyst	2 (9.52%)	1 (5.56%)	3 (5.13%)	0.64
Cyst infection	1 (4.76%)	1 (5.56%)	2 (5.14%)	0.91
30-day mortality	0 (0%)	0 (0%)	0 (0%)	1
Reintervention	3 (14.29%)	2 (11.11%)	5 (12.89%)	0.76
Rehospitalization	5 (23.81%)	2 (11.11%)	7 (17.96%)	0.30
Follow up				
Therapeutic success	18 (85.71%)	18 (100%)	35 (89.74%)	0.095
Cyst recurrence	4 (19.05%)	0 (0%)	4 (10.26%)	0.05
Median time of follow-up (days)	268 IQR 430 (46–1108)	317 IQR 984 (31–2027)	280 IQR 551 (31–2027)	0.34
Time of the stent removal (days)	106 IQR 89 (43–660)	34 IQR 24.5 (15–355)	45 IQR 68 (14–660)	0.001

Abbreviations: DPPSs, double plastic pigtail stents; LAMSs, lumen-apposing fully covered metal stents; IQR—Interquartile Range.

**Table 4 life-14-01419-t004:** Univariate analysis of Cox proportional hazard regression model for cyst recurrence.

	Univariate Analysis		
Variable	HR	95% CI	*p*
Age	1.08	0.98–1.19	0.09
Gender			0.53
Male	0.53	0.07–3.82	
Female	1		
BMI	1.06	0.9–1.28	0.47
Admission mode			0.21
Emergency	3.55	0.49–25.39	
Elective	1		
Symptoms			0.68
No	0.62	0.06–5.97	
Yes	1		
Alcohol usage			0.69
No	0.66	0.09–4.76	
Yes	1		
Complications			0.01
Yes	21.55	1.95–237.72	
No	1		
Cyst Size	1.01	0.98–1.04	0.39

Abbreviations: HR = Hazard Ratio, 95% CI = 95% Confidence Interval, BMI = Body Mass Index.

## Data Availability

The raw data supporting the conclusions of this article will be made available by the authors on request.

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
