# Peer review of "What We Know So Far About ECG for Pancreatic Pseudocysts"

_life, 2024, doi:10.3390/life14111419_

Round 1

Reviewer 1 Report

Comments and Suggestions for Authors

This is a well written and illustrated article outlining in some detail the recent experience of a high volume unit with the management of Pancreatic Pseudocysts by endoscopic means.

It does not give ANY meaningful information about the relative merits of the 2 techniques used because of scientific shortcomings. Firstly it is a retrospective review and not an RCT and secondly the technicians involved obviously used their judgement in selecting the relevant device for the clinical situation. As such no conclusions can be drawn of a comparative nature as is evidenced by the conclusion.

This needs to be retitled and seen for what it actually is, a narrative report of the experience of a high volume centre managing this disease expertly and with judgement. Any attempt to compare the stent performance in any scientific way would necessitate a prospective RCT for which this provides a reasonable platform and as the article suggests.

I would suggest the article be retitled 'What we know so far about ECG for Pancreatic Pseudocysts' and they focus on the details of the decision making of the endoscopist in their report so we can all learn how to select the best stent for the situation.

Author Response

Dear Reviewer,

Thank you for peer reviewing of our manuscript life-3237684, entitled " Plastic versus Metal Stents in the Endoscopic Cystogastrostomy for Pancreatic Pseudocysts”. 

Thank you for your questions and comments. We have fully addressed all the comments and myresponses appear below. Our revised work includes corrections according to reviewers’ comments inthe text. The changes, made according to reviewers’ comments, are marked using the red font in the main manuscript.

We take this opportunity to express my gratitude to the reviewers for their constructive and useful remarks. Their comments allowed us to identify areas in our manuscript that needed modification.

We also thank you for allowing me to resubmit a revised copy of the manuscript.

We hope that the revised manuscript is now acceptable for publication in Life.

Yours sincerely,

Beata Jabłońska.

Responses to Reviewer 1

Comment:

This is a well written and illustrated article outlining in some detail the recent experience of a high volume unit with the management of Pancreatic Pseudocysts by endoscopic means.

It does not give ANY meaningful information about the relative merits of the 2 techniques used because of scientific shortcomings. Firstly it is a retrospective review and not an RCT and secondly the technicians involved obviously used their judgement in selecting the relevant device for the clinical situation. As such no conclusions can be drawn of a comparative nature as is evidenced by the conclusion.

This needs to be retitled and seen for what it actually is, a narrative report of the experience of a high volume centre managing this disease expertly and with judgement. Any attempt to compare the stent performance in any scientific way would necessitate a prospective RCT for which this provides a reasonable platform and as the article suggests.

I would suggest the article be retitled 'What we know so far about ECG for Pancreatic Pseudocysts' and they focus on the details of the decision making of the endoscopist in their report so we can all learn how to select the best stent for the situation.

Answer:

Thank you for your positive feedback and your valuable comments. As you have pointed, the aim of this study was to show the experience of a high volume pancreatic center in the endoscopic treatment of pancreatic pseudocysts. Our results add to the current literature additional data to already existing and may be potentially used in a systematic review. The retrospective design of our study and the need to conduct a further prospective, randomized, multicenter study has been indicated in a paragraph 4.1.

4.1. Limitations

It is a retrospective analysis conducted on a selected cohort of patients from a single high volume center. Another limitation is a relatively small sample size. In addition, the use of plastic or metal stents was determined by the endoscopist. Additionally, the restricted access to hospitals during the COVID-19 pandemic could have affected the timing of prosthesis removal, potentially distorting the data on prosthesis retention times. Lastly, our follow-up period was limited, preventing us from fully capturing longer-term outcomes related to prosthesis retention and patient health. To address these limitations, future studies should consider a prospective, randomized, multicenter design with larger, more diverse patient populations.

According to your suggestions, we have changed the title as follows: 'What we know so far about ECG for Pancreatic Pseudocysts' and focused more on the clinical aspect regarding decision-making on the selection of the right stent for the patient and his clinical condition.

The following comments have been added:

The choice of the type of prosthesis depends mainly on the presence of solid masses inside the fluid collection. The second important criterion is the presence of a fistula between the collection and the pancreatic duct. In the first case, the placement of a metal stent allows for more effective drainage. The presence of a pancreatic fistula may require long-term maintenance of a drainage prosthesis, which in the case of metal stent is associated with a higher risk of bleeding [20].

We have also added additional information to our manuscript as follows:

The effectiveness of drainage is comparable in both stents. Patients treated with LAMS have fewer complications rate. Bleeding (including live threatening) is more common in this group of patients. In the group of patients treated with DPT, stent occlusion happens more often which is associated with a higher risk of infectious complications and the need to replace or add an additional stent. Migration of the metal stent is no longer a problem since the introduction of metal, self-expanding, fully covered LAMS prostheses. Migration of the DPT plastic stent occurs slightly more often than the LAMS [42]. The choice of prosthesis is also influenced by the number of drained fluid collections. In patients with two or more, it seems reasonable to deploy DPT plastic stent into each of them. Insertion of several metal prostheses could be too traumatic, as well as very expensive. It must be admitted that, considering the high and comparable drainage effectiveness of the LAMS and DPD techniques, the choice of one of them is often a subjective decision of the endoscopist.

Kind regards,

Beata Jabłońska

Reviewer 2 Report

Comments and Suggestions for Authors

The study addresses a classic topic, the endoscopic treatment of pancreatic pseudocysts, comparing plastic and metal stents.

The final conclusion is that both treatments offer similar efficacy.

There are some aspects that need to be improved:

Line 12: delete the term pancreatocystogastrostomy. The procedure must be named as endoscopic cysto-gastrostomy (ECG).

Line 13: delete the term postinflammatoy cyst. The correct term should be pancreatic pseudocyst (PPC)

Line 62: delete the term pancreatocystogastrostomy. The procedure must be named as endoscopic cysto-gastrostomy (ECG).

Line 67 after (LAMS), these are a type of self-expanding metal stents

Line 79: correct Deparment of Digestiive Tract Surgery

Line 106 delete the term pancreatocystogastrostomy. The procedure must be named as endoscopic cysto-gastrostomy (ECG).

Line 110  Change the endosonograph for the endoscopic ultrasound scope

Line 113 Delete cystoenterostomy needle knife

Line 121 When speaking of LAMS size, indicate length, diameter and the  flange diameter

Line 363 Please clarify LAMS and pigtail drainage.  Drainage of the pseudocyst is through LAMs diameter, but for pigtail stents drainage occurs mainly through the orifice produced by the two pigatils NOT through the pigtail stent itself.

Comments on the Quality of English Language

Minor English editing required

Author Response

Dear Reviewer,

Thank you for peer reviewing of our manuscript life-3237684, entitled " Plastic versus Metal Stents in the Endoscopic Cystogastrostomy for Pancreatic Pseudocysts”. 

Thank you for your questions and comments. We have fully addressed all the comments and myresponses appear below. Our revised work includes corrections according to reviewers’ comments inthe text. The changes, made according to reviewers’ comments, are marked using the red font in the main manuscript.

We take this opportunity to express my gratitude to the reviewers for their constructive and useful remarks. Their comments allowed us to identify areas in our manuscript that needed modification.

We also thank you for allowing me to resubmit a revised copy of the manuscript.

We hope that the revised manuscript is now acceptable for publication in Life.

Yours sincerely,

Beata Jabłońska.

Responses to Reviewer 2

Comment:

 The study addresses a classic topic, the endoscopic treatment of pancreatic pseudocysts, comparing plastic and metal stents.

The final conclusion is that both treatments offer similar efficacy.

There are some aspects that need to be improved:

Answer:

Thank you for your positive feedback and your valuable comments. We have fully adressed your suggestions.

Comment:

Line 12: delete the term pancreatocystogastrostomy. The procedure must be named as endoscopic cysto-gastrostomy (ECG).

Answer:

We have corrected it.                                                                                                                     

Comment:

Line 13: delete the term postinflammatoy cyst. The correct term should be pancreatic pseudocyst (PPC).

Answer:

We have corrected it.

Comment:

Line 62: delete the term pancreatocystogastrostomy. The procedure must be named as endoscopic cysto-gastrostomy (ECG).

Answer:

We have corrected it.

Comment:

Line 67 after (LAMS), these are a type of self-expanding metal stents.

Answer:

We have applied all the corrections as indicated. 

Two types of stents are currently employed: double pigtail plastic stents (DPPS) and lumen-apposing fully covered metal stents (LAMS), a type of self-expanding metal stents (SEMS) [20].

LAMS – is a type of metal, fully covered, removable, self-expandable stent (self-expandable metallic stent, also called self-expandable metal stent - SEMS) with wide flanges at the ends, which limits the possibility of dislocation of the stent. In our Endoscopy Unit, we use Olympus HANAROSTENT® Plumber™ with a diameter of 14 or 16 mm and a length of 30 or 40 mm.

Comment:

Line 79: correct Deparment of Digestiive Tract Surgery.

Answer:

We have corrected it.

Comment:

Line 106 delete the term pancreatocystogastrostomy. The procedure must be named as endoscopic cysto-gastrostomy (ECG).

Answer:

We have corrected it.

Comment:

Line 110  Change the endosonograph for the endoscopic ultrasound scope.

Answer:

We have corrected it.

Comment:

Line 113 Delete cystoenterostomy needle knife

Answer:

We have corrected it.

Comment:

Line 121 When speaking of LAMS size, indicate length, diameter and the  flange diameter.

Answer:

We have applied all the corrections and added valuable comments as indicated. 

Initially, we used two sizes of LAMS - 14 and 16 mm in diameter and 30 and 40 mm in length. Currently, we more often use 16 mm lumen diameter stent (they facilitate necrosectomy and revision of the fluid collection) and the length of the prosthesis depends only and exclusively on the distance of the pseudocyst from the wall. In the case of plastic DPD stent, we usually use one with a diameter of 10 F (3.33 mm). The selection of the length of the DPT depends on the size of the collection.

Comment:

Line 363 Please clarify LAMS and pigtail drainage.  Drainage of the pseudocyst is through LAMs diameter, but for pigtail stents drainage occurs mainly through the orifice produced by the two pigatils NOT through the pigtail stent itself.

Answer:

We have added the comments as follows:

The drainage mechanism in the LAMS consists in creating a wide (14-16 mm) channel between the fluid collection and the stomach lumen, which allows for quick drainage of the contents and, if necessary, endoscopic necrosectomy. In the case of DPT, the mechanism is complex and initially consists in drainage through the prosthesis lumen (specifically through the main channel and small holes in the prosthesis wall), and after creating a fistula channel also around the prosthesis. Inserting two prostheses through one channel connecting the collection with the stomach lumen accelerates drainage by increasing the flow through the prostheses and around them.

Round 2

Reviewer 1 Report

Comments and Suggestions for Authors

None